# CoFiCL: Coarse-to-Fine Continual Learning with Hierarchical Knowledge

## Abstract

Vision-language models (VLMs) such as Contrastive Language-Image Pre-trained model (CLIP) show strong generalization but suffer from catastrophic forgetting in continual learning. Many existing methods only use simple category prompts like "a photo of a {class name}". This ignores the fine grained knowledge that could enrich semantic representations and support transfer. In this work, we present **CoFiCL**, a coarse to fine continual learning framework. The method separates coarse category knowledge and fine conceptual knowledge, which are naturally distinguished in the CLIP text space. The coarse path uses task specific adapters to align visual features with category prompts. The fine path introduces prototype based contrastive learning over language model generated descriptions, which capture semantic relations across tasks. At inference, the two paths are fused to produce the final prediction. Experiments on multiple benchmarks show that CoFiCL improves both forward and backward transfer. The results demonstrate that hierarchical knowledge can be effectively disentangled and used to enhance continual learning in vision language models.

## 1 Introduction

Deep learning models pre-trained on large-scale datasets have achieved remarkable performance across a wide range of tasks (Radford et al., 2021; Achiam et al., 2023; Siméoni et al., 2025). However, the real world is inherently dynamic, where data arrive in a streaming manner rather than being available all at once (Zhu et al., 2024a). Adapting models to continuously incoming data often requires retraining on the entire dataset, which is costly and may violate privacy constraints (Wang et al., 2024). Continual learning (CL) (Zhou et al., 2024a) has emerged as a promising paradigm to address this challenge by enabling models to acquire new knowledge sequentially without accessing past data. Despite notable progress, CL still suffers from catastrophic forgetting (French, 1999), where models tend to overwrite previously learned knowledge when exposed to new tasks. Addressing this challenge is essential for deploying AI systems in dynamic, real-world environments where data distributions constantly evolve.

Vision-language models (VLMs) (Alayrac et al., 2022; Li et al., 2023) such as CLIP (Radford et al., 2021), trained on massive collections of image–text pairs, exhibit strong zero-shot generalization across diverse domains. However, continual learning with VLMs is more challenging than in traditional settings because they are subject to two forms of forgetting (Zheng et al., 2023; Tang et al., 2024). On the one hand, they experience backward forgetting, where knowledge of previously learned tasks is overwritten by new ones. On the other hand, they also suffer from forward forgetting, where parts of the pre-trained knowledge are eroded during adaptation. The coexistence of these two types of forgetting significantly complicates the continual learning process in VLMs.

Most existing approaches (Wang et al., 2022a; Zheng et al., 2023; Tang et al., 2024; Yu et al., 2024) on the text side of VLMs often rely on coarse prompts based on category names, such as "a photo of a leopard". This practice overlooks the fine-grained semantic concepts embedded within category names. For instance, "leopard" is associated with concepts like "muscular body" and "spotted coat". Prior studies in both neuroscience (Patterson et al., 2007; Ralph et al., 2017) and deep learning (Menon & Vondrick, 2022; Zhu et al., 2024b) suggest that such fine-grained concepts exhibit stronger transferability and generalization. For example, "lion" also shares the concept of "muscular body". Recent works (Wang et al., 2023; Zhou et al., 2025) have attempted to incorporate

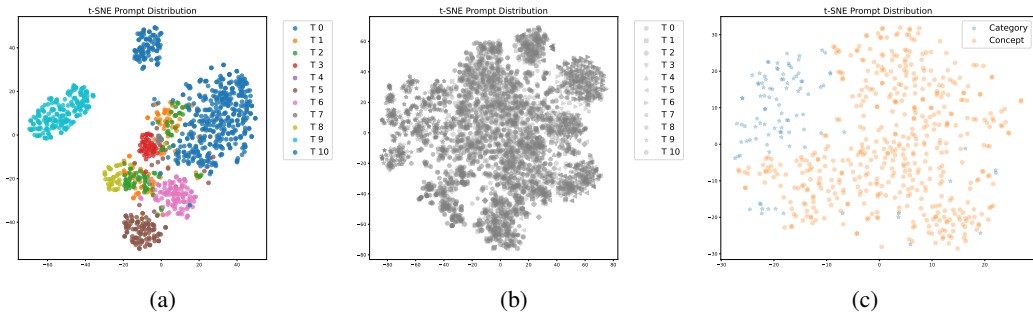

Figure 1: t-SNE visualization of prompt embeddings in the CLIP text space. (a) Category prompts capture coarse knowledge and show clear task separation. (b) Concept prompts capture fine knowledge and form a shared semantic space across tasks. (c) Within a single task, category prompts (stars) and concept prompts (circles) remain separated but complementary, which highlights the value of combining both levels of knowledge.

fine-grained knowledge into the text encoder, yet they usually lack a careful analysis of semantic structures and tend to couple coarse and fine knowledge together, which may cause interference and limit effective knowledge transfer across tasks.

Our motivation is inspired by the observations in Figure 1. Coarse prompts in Figure 1a correspond to category templates such as "`a photo of a {class name}`". They provide strong task discrimination and separate categories clearly across tasks. Fine prompts in Figure 1b are generated by a Large Language Model (LLM). They capture conceptual relations that extend across tasks and support transfer. Within a single task, as shown in Figure 1c, category prompts and concept prompts remain well separated while encoding complementary aspects of semantics. This indicates that coarse and fine knowledge can be modeled jointly.

Based on this, we propose **CoFiCL (Coarse-to-Fine Continual Learning)**, a dual path framework that models hierarchical knowledge explicitly. We place lightweight adapters (Chen et al., 2022) in both the visual and the text encoders. This allows the two branches to adapt while preserving the general representation space of CLIP. In the **coarse path**, task specific adapters align visual and textual features with category prompts such as "`a photo of a {class name}`". Training is guided by cross entropy loss, and Gaussian centers built from features are used for adapter selection at inference. In the **fine path**, the large language model generates multiple concept descriptions without explicit class names, for example "`It is {desc}`". These prompts are only applied on the visual side. A prompt pool (Jia et al., 2022; Wang et al., 2022c) is maintained, and the selected prompts are prepended to visual embeddings. The resulting features are trained with prototype based contrastive learning to capture relations across tasks. During inference, an image query interacts with Gaussian centers in the coarse path to select adapters, and with semantic centers in the fine path to retrieve prompts. The outputs of the two paths are fused, which produces predictions that are both discriminative and transferable. CoFiCL therefore disentangles hierarchical knowledge in continual learning and combines the strengths of category prompts and conceptual prompts.

- We propose CoFiCL, a dual-path framework that disentangles coarse and fine knowledge to alleviate both forward and backward forgetting in continual learning of VLMs.

- The coarse path introduces task specific adapters guided by category prompts, while the fine path relies on LLM-generated descriptions and prototype based contrastive learning to enhance transfer.

- Experiments on multiple benchmarks show that CoFiCL achieves state of the art results in knowledge retention and knowledge transfer.

## 2 METHODOLOGY

### 2.1 PRELIMINARIES

Traditional studies in continual learning often construct task sequences by splitting the classes of a single dataset into disjoint subsets. While such setups are convenient for controlled comparisons, they remain far from realistic scenarios, where new tasks typically involve both domain shifts and novel classes emerging over time. In this work, we consider the more realistic and challenging setting of domain-class incremental learning (DCIL) (Zheng et al., 2023; Tang et al., 2024; Xu et al., 2024). In DCIL, a model receives a stream of tasks $\{D^1, D^2, \ldots, D^T\}$, where each task $D^t = \{(x_i^t, y_i^t)\}_{i=1}^{N_t}$ contains $N_t$ instances and a disjoint class set $\mathcal{C}^t$, i.e., $\mathcal{C}^t \cap \mathcal{C}^{t'} = \phi$ and $\mathbb{P}^t \neq \mathbb{P}^{t'}$ for $t \neq t'$, where $\mathbb{P}^t$ denotes the data distribution of task $t$.

We adopt a pretrained CLIP model $f = (f_i, f_t)$ consisting of an image encoder $f_i$ and a text encoder $f_t$. Given an image $x$ and a text prompt $t_c$ for class $c$, their embeddings are

$$z_i = f_i(x), \qquad z_t = f_t(t_c), \tag{1}$$

where $z_i, z_t \in \mathbb{R}^d$ share the same embedding dimension. The similarity is measured by cosine similarity, and the class probability is obtained via a temperature-scaled softmax:

$$s_c(x) = \frac{\exp(\cos(z_i, z_t)/\tau)}{\sum_{c' \in \mathcal{C}^{(t)}} \exp(\cos(z_i, f_t(t_{c'}))/\tau)}, \tag{2}$$

where $\tau$ is a temperature parameter.

### 2.2 COARSE-TO-FINE PROMPT CONSTRUCTION

Given a class label $c \in \mathcal{C}^t$ from task $t$, CoFiCL constructs both coarse-grained prompts and fine-grained prompts to capture complementary semantics (see Figure 2(a)).

**Coarse-grained prompts.** Following prior works (Radford et al., 2021; Zheng et al., 2023), we adopt category-level templates to construct coarse-grained descriptions. For each class $c$, we instantiate a prompt

$$t_c^{\text{coarse}} = \texttt{a photo of a \{class name\}}, \tag{3}$$

where the template may vary slightly across datasets, as detailed in Appendix. These coarse prompts provide strong task discriminability and serve as global semantic anchors by aligning visual features with explicit category names.

**Fine-grained prompts.** While coarse prompts emphasize categorical identity, they fail to capture transferable semantic relations. Inspired by recent works on attribute-augmented prompts (Zhu et al., 2024b), we construct fine-grained prompts via LLMs. Specifically, for each class $c$ from dataset $D^t$, we form an instruction by combining the class name with the dataset description and feed it into GPT-4o (Achiam et al., 2023) through its API. The LLM generates a set of fine-grained concept descriptions:

$$\mathcal{D}_c = \{d_{c,1}, d_{c,2}, \ldots, d_{c,K}\}, \tag{4}$$

where each $d_{c,k}$ corresponds to a transferable attribute or property of the class (e.g., "muscular body", "spotted coat" for "leopard"). To avoid direct class-name leakage, we reformat each description into the template

$$t_{c,k}^{\text{fine}} = \texttt{It is \{desc\}}, \qquad d_{c,k} \in \mathcal{D}_c, \tag{5}$$

yielding prompts that focus on class-agnostic semantics. This design encourages the model to capture relational knowledge across tasks, as different categories may share overlapping concepts (e.g., both "lion" and "leopard" share "muscular body"). We provide the full LLM prompt design as well as qualitative examples of generated descriptions in Appendix.

By disentangling prompt construction into coarse and fine branches, CoFiCL ensures that category-level discriminability and transferable conceptual knowledge are explicitly encoded, forming the foundation for our dual-path continual learning framework.

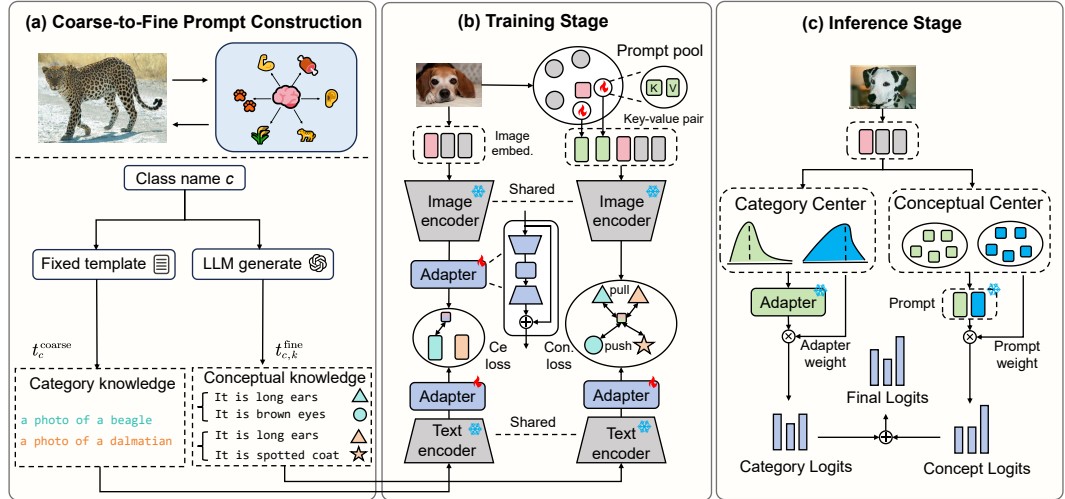

Figure 2: Overview of CoFiCL. (a) Prompt construction from category names and LLM descriptions. (b) Training with task adapters in the coarse path and prototype learning in the fine path. (c) Inference by retrieving adapters and prompts via centers and fusing both paths for prediction.

## 2.3 COARSE-TO-FINE CONTINUAL LEARNING

Figure 2 gives an overview of CoFiCL. The coarse to fine framework models hierarchical knowledge with two complementary branches. We place lightweight adapters in both the visual encoder and the text encoder. Adapters let the model adapt to new tasks while keeping the main pretrained representations stable. The fine path uses a prompt pool that is applied on the visual side only. The two branches focus on different goals and are trained jointly.

**Coarse grained knowledge learning.** The coarse branch captures category level discriminative knowledge by aligning adapted features with explicit class names. We insert adapters $\mathcal{A}(\cdot)$ into the image encoder and into the text encoder. Given input features $\boldsymbol{X} \in \mathbb{R}^{n \times d}$, an adapter produces

$$\boldsymbol{Y} = \mathcal{A}(\boldsymbol{X}) = s \cdot \sigma(\boldsymbol{X}\boldsymbol{W}_{\text{down}})\boldsymbol{W}_{\text{up}} + \boldsymbol{X}, \tag{6}$$

where $\boldsymbol{W}_{\text{down}} \in \mathbb{R}^{d \times r}$ and $\boldsymbol{W}_{\text{up}} \in \mathbb{R}^{r \times d}$, $\sigma(\cdot)$ is a nonlinearity, and $s$ is a scalar. We apply the same form to adapters in both encoders. After adapter modulation, the [CLS] token is used as the feature for classification.

To enable adapter selection across tasks, we record Gaussian task centers Tang et al. (2024), from the raw visual features. For task $i$ we compute

$$\mu_i = \mathbb{E}_{x_j \sim D^i}[f_i(x_j)], \qquad \Sigma_i = \mathbb{E}_{x_j \sim D^i}[(f_i(x_j) - \mu_i)^{\top}(f_i(x_j) - \mu_i)], \tag{7}$$

where $\mu_i$ and $\Sigma_i$ are the mean and covariance of task features. These centers guide adapter selection at test time. During training, adapter modulated features are paired with category text embeddings $f_t(t_c^{\text{coarse}})$ and trained with cross entropy loss. The coarse classification loss is

$$\mathcal{L}_{\text{coarse}} = - \sum_{c \in \mathcal{C}^t} y_c \log s_c(x). \tag{8}$$

**Fine grained knowledge learning.** The fine branch captures transferable semantic concepts. We keep a prompt pool $\mathcal{P} = \{(k_m, v_m)\}_{m=1}^M$, where $k_m$ is a key and $v_m$ is the prompt value. Given an image, its [CLS] feature $z_{\text{cls}}$ queries the pool. We compute cosine similarities and obtain per prompt probabilities

$$p_m(z_{\text{cls}}) = \frac{\exp(\cos(z_{\text{cls}}, k_m)/\tau_p)}{\sum_{n=1}^M \exp(\cos(z_{\text{cls}}, k_n)/\tau_p)}. \tag{9}$$

We select the top $K$ prompts according to these scores and prepend their values before the [CLS] token. The prompt modulated feature is

$$\tilde{z}_i = f_i([v_{m_1}, \ldots, v_{m_K}, \text{CLS}, \ldots]). \tag{10}$$

To avoid prompt over concentration, we maintain a running selection count $C_m$ for each prompt and normalize it to a frequency $f_m = C_m / \sum_n C_n$.

We build concept prototypes by clustering the fine prompt text embeddings $\{f_t(t_{c,k}^{\text{fine}})\}$ with $k$-means. Let $\{p_n\}_{n=1}^N$ be the resulting prototypes. The prompt modulated image features are aligned to prototypes with a contrastive loss:

$$\mathcal{L}_{\text{proto}} = -\log \frac{\exp(\cos(\tilde{z}_i, p^+)/\tau)}{\sum_{p \in \mathcal{P}} \exp(\cos(\tilde{z}_i, p)/\tau)}, \tag{11}$$

where $p^+$ is the assigned prototype for the sample.

We also add a simple prompt similarity term to measure alignment between queries and prompts. Let $p_m$ denote the average similarity-based probability for prompt $m$ on the current batch, and let $f_m$ be the normalized historical frequency. We define

$$\mathcal{L}_{\text{sim}} = -\sum_{m=1}^M f_m\, p_m. \tag{12}$$

Minimizing this term encourages the model to select prompts that best match the current queries.

**Overall objective.** The full training loss combines the three parts:

$$\mathcal{L} = \lambda_{\text{coarse}} \mathcal{L}_{\text{coarse}} + \lambda_{\text{proto}} \mathcal{L}_{\text{proto}} + \lambda_{\text{sim}} \mathcal{L}_{\text{sim}}, \tag{13}$$

where $\lambda_{\text{coarse}}, \lambda_{\text{proto}}, \lambda_{\text{sim}}$ are weights. During training we update adapter parameters, prompt keys and values, prototype centers, and the prompt counts $C_m$.

**Inference.** At test time an image query interacts with both Gaussian task centers and concept centers. For adapter selection we compute the log likelihood of the query under each task Gaussian:

$$S^i = \log \varphi(x_q; \mu^i, \Sigma^i), \tag{14}$$

where $\varphi(\cdot; \mu, \Sigma)$ is the multivariate normal density. We standardize the scores across tasks to obtain $\hat{S}^i$, for example by subtracting the mean and dividing by the standard deviation of $\{S^j\}_j$. We then map the standardized score to a scaling weight with a sigmoid:

$$\mathcal{W}(\hat{S}^i) = \frac{1}{1 + \exp(-\hat{S}^i)}. \tag{15}$$

We use these weights to combine coarse logits from different task adapters. Concretely, if $s_i^{\text{coarse}}(x)$ is the coarse logits produced by adapter $i$, we form

$$s^{\text{coarse}}(x) = \sum_i \mathcal{W}(\hat{S}^i)\, s_i^{\text{coarse}}(x). \tag{16}$$

Prompt weights are taken from the query key similarities. We compute $p_m(z_{\text{cls}})$ as in training and use them to weight fine logits. If $s_m^{\text{fine}}(x)$ is the fine logits associated with prompt $m$, we form

$$s^{\text{fine}}(x) = \sum_m p_m(z_{\text{cls}})\, s_m^{\text{fine}}(x). \tag{17}$$

The two paths are fused by a final weighted sum

$$s(x) = \alpha \cdot s^{\text{coarse}}(x) + (1 - \alpha) \cdot s^{\text{fine}}(x), \tag{18}$$

where $\alpha$ controls the balance between discriminative category knowledge and transferable conceptual knowledge.

This design lets each query search over past tasks and retrieve both global category knowledge and local concept knowledge. The fused output combines both strengths and yields robust classification in continual learning.

## 3 EXPERIMENTS

### 3.1 IMPLEMENTATION DETAILS

**Benchmarks.** We evaluate our method on the MTIL benchmark under the exemplar-free protocol, where no past instances are stored. MTIL (Zheng et al., 2023): built from 11 datasets across diverse domains, including Aircraft (Maji et al., 2013), Caltech101 (Fei-Fei et al., 2004), CIFAR100 (Krizhevsky et al., 2009), DTD (Cimpoi et al., 2014), EuroSAT (Helber et al., 2019), Flowers (Nilsback & Zisserman, 2008), Food (Bossard et al., 2014), MNIST (Deng, 2012), Oxford-Pet (Parkhi et al., 2012), StanfordCars (Krause et al., 2013), and SUN397 (Xiao et al., 2010). It covers over 1,201 classes in total. We report results under Order I, Order II, and Few-Shot (FS) settings.

**Comparison Methods.** We compare our approach with SOTA continual learning methods. Full fine-tuning methods: LwF (Li & Hoiem, 2017), iCaRL (Rebuffi et al., 2017), ZSCL (Zheng et al., 2023). Parameter-efficient methods: L2P (Wang et al., 2022c), DualPrompt (Wang et al., 2022b), S-Prompt (Wang et al., 2022a), MoE-Adapter (Yu et al., 2024), DIKI (Tang et al., 2024). All methods share the same pre-trained CLIP ViT-B/16 backbone and incremental learning protocol for fair comparison.

**Training Details.** We use CLIP ViT-B/16 as the backbone, inserting adapters with a bottleneck dimension of 64 and scalar 1.0 in every visual layer. Each task has a pool of 32 prompts of 4 tokens, and the 2 most relevant prompts are selected per instance. For fine-grained concept knowledge, GPT-4o (Achiam et al., 2023) generates 5 concepts per class, and prototypes are clustered with $K_{\text{num}} = [100, 100, 100, 47, 10, 102, 101, 10, 37, 196, 397]$. Training uses SGD with learning rate 0.1, batch size 64, for 10 epochs. Loss weights are $\lambda_{\text{coarse}} = 1.0$, $\lambda_{\text{proto}} = 0.1$, $\lambda_{\text{sim}} = 0.1$. At test time, category and concept logits are fused with weights 0.8 and 0.2. All experiments run on a single NVIDIA A100 80G GPU.

**Evaluation Metrics.** We use three metrics to evaluate continual learning performance: (i) *Transfer*, measuring zero-shot generalization to unseen tasks; (ii) *Last*, the accuracy on the first task after learning all tasks, which reflects backward forgetting; and (iii) *Average*, the mean accuracy across all tasks. Higher values indicate better knowledge retention and transfer. Detailed formulas are provided in the Appendix.

### 3.2 BENCHMARK COMPARISON

We first evaluate CoFiCL on the MTIL benchmark under Order I, with detailed results reported in Table 1. CoFiCL consistently outperforms all baselines across most datasets, yielding overall gains of **+1.5%** (Transfer), **+1.6%** (Average), and **+1.4%** (Last) compared with the strongest prior method. In particular, CoFiCL achieves stable improvements on both fine-grained datasets (e.g., DTD, Cars) and large-scale ones (e.g., SUN397), confirming its ability to balance task discriminability and cross-task generalization. The advantages of CoFiCL mainly stem from its dual-path design. The *coarse-grained path* leverages task-specific adapters to preserve category-level discriminative power, while the *fine-grained path* employs LLM-generated concepts and prototype-based contrastive learning to capture semantic relations beyond single tasks. This separation reduces interference between the two types of knowledge and enables complementary learning.

We further examine robustness on MTIL with Order II (Table 2(a)) and the few-shot setting MTIL-FS (Table 2(b)). On Order II, CoFiCL achieves average improvements of **+1.9%** (Transfer), **+1.2%** (Average). On MTIL-FS, the gains reach **+2.1%**, **+1.3%**, and **+0.9%**, respectively. These results highlight that CoFiCL adapts well to task order variations and limited data, providing both strong retention of past knowledge and effective forward transfer.

### 3.3 FURTHER ANALYSIS

**Ablation Study.** Table 3 reports the results of ablation experiments. Using only category logits (Line 2) or only prompt logits (Line 5) leads to inferior performance. The combination of both achieves the best results. For category logits, replacing task-specific Gaussian centers with a random

Table 1: *Transfer*, *Average*, and *Last* scores of different continual learning methods on MTIL benchmark with Order-I.

| Method | Aircraft | Caltech101 | CIFAR100 | DTD | EuroSAT | Flowers | Food | MNIST | OxfordPet | Cars | SUN397 | Average |
|---|---|---|---|---|---|---|---|---|---|---|---|---|
| Zero-shot | 24.8 | 92.9 | 68.4 | 43.8 | 47.7 | 71.4 | 85.8 | 59.5 | 89.1 | 65.8 | 62.6 | 64.7 |
| Upper Bound | 62.0 | 96.2 | 89.6 | 79.5 | 98.9 | 97.5 | 92.7 | 99.6 | 94.7 | 89.6 | 81.8 | 89.3 |
| **Transfer** | | | | | | | | | | | | |
| LwF | | 74.5 | 56.9 | 39.1 | **51.1** | 52.6 | 72.8 | 60.6 | 75.1 | 30.3 | 55.9 | 56.9 |
| iCaRL | | 56.6 | 44.6 | 32.7 | 39.3 | 46.6 | 68.0 | 46.0 | 77.4 | 31.9 | 60.5 | 50.4 |
| ZSCL | | 86.0 | 67.4 | 45.4 | 50.4 | 69.1 | 87.6 | 61.8 | 86.8 | 60.1 | **66.8** | 68.1 |
| L2P | | 65.6 | 50.9 | 30.4 | 41.4 | 49.3 | 71.8 | 36.3 | 77.5 | 55.3 | 53.4 | 53.2 |
| DualPrompt | | 56.7 | 51.4 | 28.7 | 33.7 | 45.6 | 70.9 | 59.5 | 77.7 | 49.5 | 50.4 | 52.4 |
| S-Prompts | | 67.3 | 49.4 | 26.4 | 39.7 | 47.1 | 70.2 | 34.3 | 78.9 | 56.7 | 52.2 | 52.2 |
| MoE-Adapter | | 87.9 | 68.2 | 44.4 | 49.9 | **70.7** | **88.7** | 59.7 | **89.1** | 64.5 | 65.5 | 68.9 |
| DIKI | | **92.9** | 69.0 | 43.2 | 48.2 | 67.4 | 85.2 | 63.0 | 87.9 | 63.8 | 66.2 | 68.7 |
| CoFiCL | | 90.4 | **69.1** | **51.7** | 50.3 | 69.1 | 85.6 | **63.3** | 89.0 | 66.3 | 66.8 | **70.2** |
| **Average** | | | | | | | | | | | | |
| LwF | 36.3 | 86.9 | 72.0 | 59.0 | 73.7 | 60.0 | 73.6 | 74.8 | 80.0 | 37.3 | 58.1 | 64.7 |
| iCaRL | 35.5 | 89.2 | 72.2 | 60.6 | 68.8 | 70.0 | 62.3 | 81.8 | 41.2 | 62.5 | 65.7 | |
| ZSCL | 45.1 | 92.0 | 80.1 | 64.3 | 79.5 | 81.6 | **89.6** | 75.2 | 88.9 | 64.7 | **68.0** | 75.4 |
| L2P | 38.0 | 85.2 | 78.2 | 61.3 | 72.9 | 74.9 | 79.7 | 59.1 | 82.0 | 59.7 | 55.4 | 67.9 |
| DualPrompt | 37.8 | 84.3 | 78.6 | 60.1 | 71.1 | 73.2 | 79.1 | 73.9 | 82.3 | 55.1 | 52.8 | 68.0 |
| S-Prompts | 37.5 | 92.5 | 77.5 | 58.2 | 76.4 | 74.1 | 78.8 | 57.9 | 83.0 | 60.8 | 54.4 | 68.3 |
| MoE-Adapter | 50.2 | 91.9 | 83.1 | **69.4** | 78.9 | 84.0 | 89.1 | 73.7 | 89.3 | 67.7 | 66.9 | 76.7 |
| DIKI | 45.1 | 95.5 | 83.1 | 64.8 | 79.9 | 83.5 | 87.0 | 76.2 | 89.6 | 67.0 | 67.1 | 76.3 |
| CoFiCL | **51.7** | **95.9** | **84.9** | 67.3 | **80.8** | **84.9** | 87.1 | **76.4** | 90.4 | 69.5 | 67.8 | **77.9** |
| **Last** | | | | | | | | | | | | |
| LwF | 26.3 | 87.5 | 71.9 | 66.6 | 79.9 | 66.9 | 83.8 | **99.6** | 92.1 | 66.1 | 80.4 | 74.6 |
| iCaRL | 35.8 | 93.0 | 77.0 | 70.2 | 83.3 | 88.5 | 90.4 | 86.7 | 93.2 | 81.2 | **81.9** | 80.1 |
| ZSCL | 40.6 | 92.2 | 81.3 | 70.5 | 94.8 | 90.5 | **91.9** | 98.7 | 93.9 | 85.3 | 80.2 | 83.6 |
| L2P | 38.0 | 87.1 | 84.2 | 72.9 | 86.0 | 96.1 | 89.2 | 99.0 | 94.1 | 79.6 | 76.0 | 82.0 |
| DualPrompt | 37.8 | 87.1 | 84.6 | 71.8 | 89.2 | 96.3 | 89.1 | 99.1 | 94.5 | 79.9 | 76.5 | 82.3 |
| S-Prompts | 37.5 | 95.1 | 83.7 | 70.2 | 97.5 | 96.5 | 89.0 | 99.1 | 94.0 | 79.5 | 75.8 | 83.4 |
| MoE-Adapter | 49.8 | 92.2 | 86.1 | **78.1** | 95.7 | 94.3 | 89.5 | 98.1 | 89.9 | 81.6 | 80.0 | 85.0 |
| DIKI | 45.2 | 95.7 | 86.3 | 72.9 | 98.0 | 97.0 | 89.2 | 99.4 | 94.2 | 81.6 | 76.6 | 85.1 |
| CoFiCL | **51.8** | **96.4** | **88.4** | 73.0 | **98.3** | **98.1** | 89.0 | 99.4 | **94.4** | 84.4 | 78.4 | **86.5** |

Table 2: Performance comparison on MTIL under different settings.

(a) Order II

| Method | Transfer | Average | Last | Params. |
|---|---|---|---|---|
| LwF | 53.2 | 62.2 | 71.9 | 211M |
| iCaRL | 50.9 | 56.9 | 71.6 | 211M |
| ZSCL | 64.2 | 74.5 | 83.4 | 211M |
| L2P | 42.5 | 62.5 | 82.3 | 0.5M |
| DualPrompt | 52.1 | 67.5 | 82.8 | 1.8M |
| S-Prompts | 45.3 | 65.1 | 83.8 | 0.5M |
| MoE-Adapter | 64.3 | 74.7 | 84.1 | 59.8M |
| DIKI | 64.4 | 74.5 | **85.5** | 1.8M |
| CoFiCL | **66.3** | **75.7** | 85.4 | 27.3M |

(b) Few-Shot

| Method | Transfer | Average | Last |
|---|---|---|---|
| ZSCL | 68.3 | 69.3 | 74.0 |
| L2P | 53.9 | 62.3 | 73.3 |
| DualPrompt | 57.9 | 64.3 | 74.7 |
| S-Prompts | 55.5 | 63.2 | 73.8 |
| DIKI | 70.3 | 71.9 | 77.1 |
| CoFiCL | **72.4** | **73.2** | **78.0** |

adapter causes a large drop (Line 3). Removing adapter weights also hurts performance since irrelevant adapters are forced to contribute (Line 4). For prompt logits, discarding prompt weights further degrades performance (Line 6). Prompt initialization is also important. Zero initialization works best, while removing it leads to a significant decline (Line 7). The full model with all components (Line 1) consistently achieves the highest performance.

Table 3: Ablation study of CoFiCL.

| Adapter (cate. logits) | Adapter (Gau. center) | Adapter (Ada. weight) | Prompt (con. logits) | Prompt (prom. weight) | Prompt (zero init) | Trans. | Avg. | Last |
|---|---|---|---|---|---|---|---|---|
| ✓ | ✓ | ✓ | ✓ | ✓ | ✓ | 70.2 | 77.9 | 86.5 |
| ✗ | | | ✓ | ✓ | ✓ | 38.8 | 36.5 | 35.7 |
| ✓ | ✗ | | ✓ | ✓ | ✓ | 64.0 | 63.2 | 63.8 |
| ✓ | ✓ | ✗ | ✓ | ✓ | ✓ | 68.4 | 76.1 | 84.7 |
| ✓ | ✓ | ✓ | ✗ | | | 66.7 | 76.6 | 86.4 |
| ✓ | ✓ | ✓ | ✓ | ✗ | | 69.4 | 76.1 | 84.3 |
| ✓ | ✓ | ✓ | ✓ | ✓ | ✗ | 69.5 | 77.2 | 85.8 |

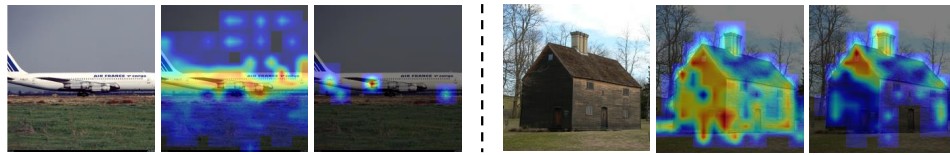

Figure 3: GradCAM visualization of adapter and prompt features for two examples. For each example, the three images from left to right show the original image, adapter features focusing global object structures, and prompt features attending to local and transferable regions.

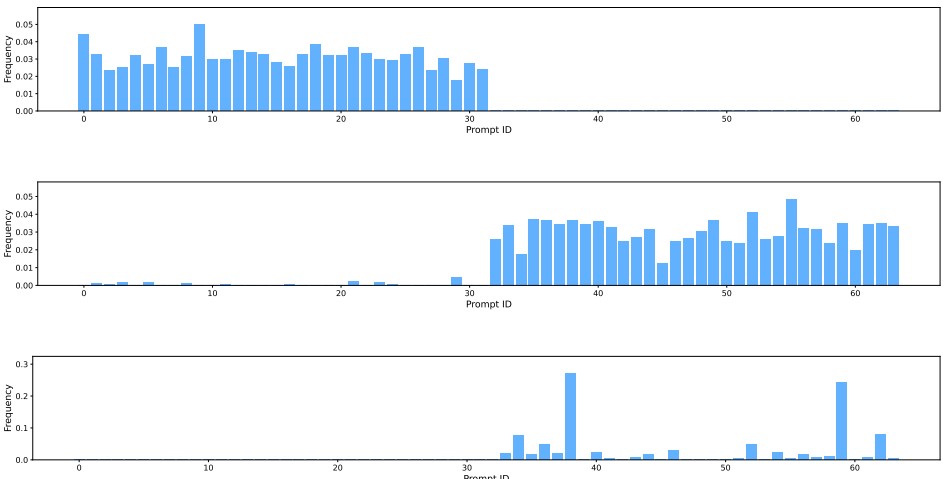

Figure 4: Prompt selection after training Task 1. Queries from Task 0, Task 1, and Task 2 show clear task relevance with visible cross-task transfer.

**GradCAM for Adapter vs Prompt.** Figure 3 visualizes the attention regions of adapters (middle) and prompts (right) using GradCAM (Selvaraju et al., 2017). Adapters highlight global structures and overall shapes, reflecting task-specific but broad patterns. Prompts, in contrast, attend to localized regions that carry transferable semantic cues. This complementary focus illustrates why their combination yields both task discrimination and cross-task generalization.

**Prompt Selection Dynamics.** Figure 4 illustrates the prompt usage after training on Task 1. When testing Task 0 (top), almost all queries select prompts from Task 0, consistent with its fine-grained nature in Aircraft. When testing Task 1 (middle), most queries select prompts from Task 1, while a small portion still relies on Task 0, indicating knowledge transfer. When testing Task 2 (bottom), the majority of queries select prompts from Task 1 due to higher similarity. This visualization shows that prompt selection captures task relevance and enables cross-task transfer.

**Prompt Feature Analysis.** Figure 5 presents three analyses of learned prompts. Subfigure (a) reports prompt diversity (Hong et al., 2025) as the number of tasks increases. We compute diversity for each prompt pool as the nuclear norm of the flattened prompt tensor, and then average across all pools. Diversity grows with more tasks, showing that prompt learning captures richer and more heterogeneous semantics. Subfigure (b) examines the sensitivity to prompt length and top-$k$ selection. We visualize the Average score on a grid, with color intensity indicating performance. The results reveal stable performance across a range of settings, suggesting robustness to hyperparameters. Subfigure (c) studies the effect of prompt pool size. Accuracy improves as the pool enlarges, but the gain becomes marginal beyond a certain size. This indicates that moderate pool sizes balance performance and efficiency.

**Case Study of Error Correction.** Figure 6 shows examples where the coarse path fails but the fine path succeeds. We display the input image with the top-5 category logits and top-5 concept logits.

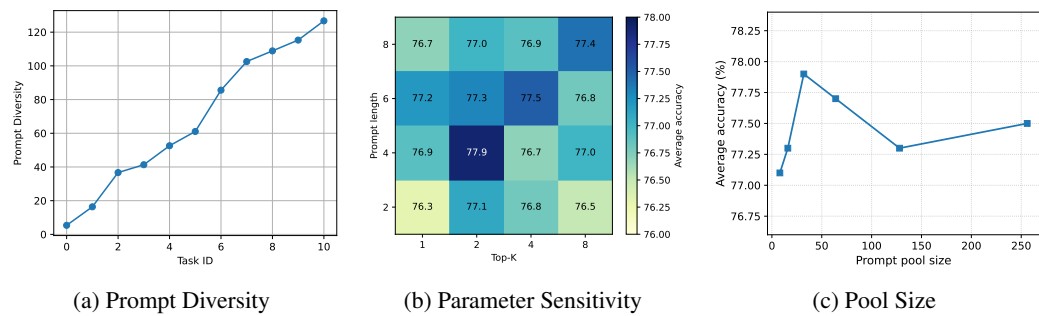

(a) Prompt Diversity        (b) Parameter Sensitivity        (c) Pool Size

Figure 5: Prompt feature analysis: (a) diversity increases with more tasks, (b) stable performance across prompt length and top-$k$, (c) pool size yields diminishing returns.

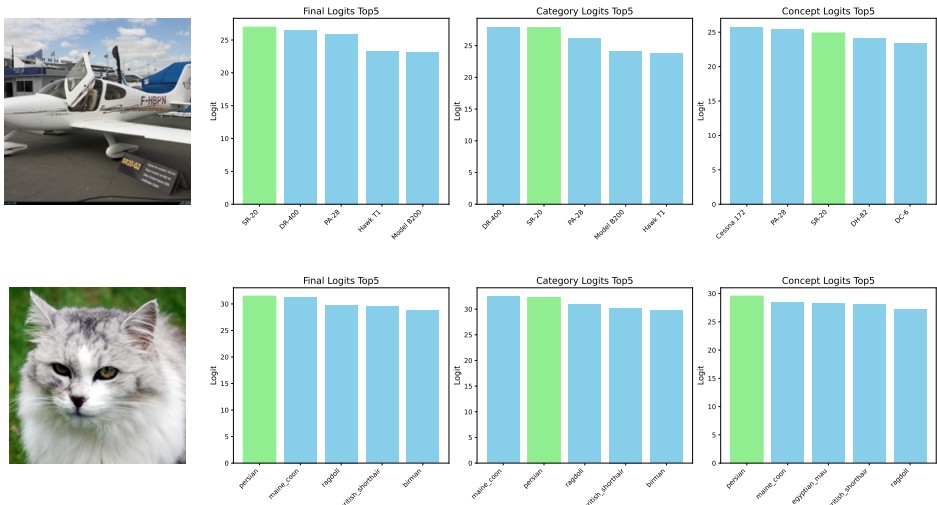

Figure 6: Case study of error correction. Examples where using only category logits leads to incorrect predictions, while incorporating concept logits corrects the final result.

In several cases, category logits predict a wrong class, while concept logits identify the correct one. The final fused prediction follows the concept path and becomes correct. GradCAM highlights that prompts attend to discriminative local cues, which explain why concept logits provide better guidance. This demonstrates that the fine path can effectively correct errors from the coarse path and improve robustness.

## 4 CONCLUSION

We propose **CoFiCL**, a coarse-to-fine continual learning framework for vision-language models. By disentangling coarse-grained category knowledge and fine-grained conceptual knowledge, CoFiCL leverages complementary semantics to improve both task discrimination and cross-task transfer. Task-specific adapters preserve category-level information, while LLM-generated prompts and prototype-based contrastive learning capture transferable concepts. Extensive experiments on multiple DCIL benchmarks demonstrate that CoFiCL achieves superior forward and backward transfer, adapts to varying task orders, and effectively mitigates catastrophic forgetting. Our analyses show that adapters focus on global structures, prompts attend to local cues, and prompt selection reflects knowledge transfer across tasks. CoFiCL provides a general and effective strategy for enhancing continual learning in vision-language models.

## 5 ETHICS STATEMENT

This work does not involve new human or animal subjects, personally identifiable information, or sensitive content. All datasets used are publicly available and commonly adopted in the vision-language and continual learning communities. We believe our method does not raise ethical concerns beyond those already present in prior research on vision-language models.

## 6 REPRODUCIBILITY STATEMENT

We have made substantial efforts to ensure the reproducibility of our results. Implementation details, including datasets, training protocols, and hyperparameter choices, are provided in Section 3.1, Appendix C, and Appendix D. Additional results and ablation studies are reported in Appendix F. All code, configuration files, and instructions for reproducing the experiments are included in the supplementary materials and will be released upon publication.

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

## A  LLM USAGE

We used large language models (LLMs) only as writing assistants to polish the presentation and improve readability. LLMs were not involved in research ideation, experimental design, implementation, or analysis. All technical contributions, experiments, and interpretations are solely the responsibility of the authors.

## B  RELATED WORK

### B.1  CONTINUAL LEARNING

Continual learning (CL) (Wang et al., 2024) aims to enable models to acquire new knowledge sequentially without forgetting what has been learned before. Existing methods are commonly categorized into three families: rehearsal-based, regularization-based, and architecture-based approaches. Rehearsal-based methods (Lopez-Paz & Ranzato, 2017; Rebuffi et al., 2017; Liang et al., 2024) mitigate forgetting by storing a small buffer of past samples or synthesizing pseudo-data for replay. While often effective, they face memory constraints and require careful sampling strategies. Regularization-based methods (Kirkpatrick et al., 2017; Zeng et al., 2019) constrain parameter updates or prediction distributions to preserve past knowledge. These approaches avoid storing old data but may struggle when tasks are long and diverse. Architecture-based methods (Yan et al., 2021; Douillard et al., 2022) dynamically allocate or isolate parameters across tasks, which can preserve task-specific knowledge but often increase model complexity or capacity demands.

### B.2  PARAMETER-EFFICIENT CONTINUAL LEARNING

Recent methods increasingly build on pre-trained models by freezing most parameters and tuning only lightweight modules (Zhou et al., 2024a). A major line of work is prompt tuning (Jia et al., 2022), which insert small trainable modules into the backbone to enable efficient task-specific adaptation. Some approaches maintain a pool of prompts (Wang et al., 2022c;b) for different tasks, while others generate prompts dynamically through attention (Smith et al., 2023) or generative mechanisms (Jung et al., 2023; Roy et al., 2024). These strategies flexibly adapt to task information with minimal parameter cost. Another line focuses on adapters (Houlsby et al., 2019; Chen et al., 2022), which insert small trainable modules into the backbone. Beyond static adapters, extensions include mixture-of-experts (Yu et al., 2024) designs for dynamic routing and subspace-expansion strategies (Zhou et al., 2024b) that allocate lightweight task-specific subspaces. Overall, these parameter-efficient fine-tuning (PEFT) approaches have become the dominant paradigm in continual learning

with pre-trained models. Among PEFT approaches for VLMs, two works are most related to ours. AttriCLIP (Wang et al., 2023) focuses on attribute modeling but does not explicitly incorporate fine-grained descriptions of categories. ENGINE (Zhou et al., 2025) introduces textual descriptions, yet it overlooks the structural distinction between coarse- and fine-grained semantic knowledge. In contrast, our CoFiCL explicitly disentangles coarse- and fine-grained knowledge into complementary paths, achieving a more balanced trade-off between knowledge retention and transfer.

## C    PROMPT CONSTRUCTION DETAILS

We provide detailed information on the prompts used in our framework across three tables. Table 4 lists the coarse-grained prompt templates designed for different datasets. These templates capture general characteristics of each dataset and serve as the starting point for generating fine-grained descriptions.

Table 5 summarizes the dataset descriptions used for constructing fine-grained prompts. Each description provides relevant context about the classes, which guides the generation of more detailed class-level prompts.

Figure 7 presents the actual prompts used to query the LLM for generating fine-grained class descriptions. These prompts combine information from the coarse templates and dataset descriptions (Zhu et al., 2024b) to produce diverse and semantically rich class-level representations.

Table 4: Coarse-grained prompt templates for different datasets.

| Dataset | Fixed template |
|---|---|
| Aircraft | *a photo of a {class name}, a type of aircraft* |
| DTD | *a photo of a {class name} texture* |
| EuroSAT | *a centered satellite photo of {class name}* |
| Flowers | *a photo of a {class name}, a type of flower* |
| Food | *a photo of a {class name}, a type of food* |
| MNIST | *a photo of the number: {class name}* |
| Oxford Pet | *a photo of a {class name}, a type of pet* |
| Stanford Cars | *a photo of a {class name}, a type of car* |
| Caltech101, CIFAR100, SUN397 | *a photo of a {class name}* |

Table 5: Dataset descriptions used for constructing fine-grained prompts.

| Dataset | Descriptions |
|---|---|
| Aircraft | *contains images of different aircraft model variants, most of which are airplanes* |
| Caltech101 | *contains images from 101 object categories* |
| CIFAR100 | *32x32 colour images in 100 classes* |
| DTD | *has collection of textural images in the wild* |
| EuroSAT | *based on Sentinel-2 satellite images for land use and land cover classification* |
| Flowers | *the flowers chosen to be flowers commonly occurring in the United Kingdom with large scale, pose, and light variations* |
| Food | *consists of 101 food categories with some amount of noise* |
| MNIST | *28x28 black-and-white images of handwritten digits extracted from two NIST databases* |
| Oxford Pet | *a pet dataset whose images have a large variation in scale, pose, and lighting* |
| Stanford Cars | *contains images of cars whose classes are typically at the level of Make, Model, Year, ex* |
| SUN397 | *a Scene UNderstanding dataset with 397 categories* |

```
You are given a class label from an image classification dataset.

Class name: "{class_name}"
Dataset description: "{dataset_desc}"

Please generate 3 to 5 concise and representative descriptions of this class.
- Each description should be concise (a short phrase or short sentence).
- Include objective characteristics (shape, texture, parts, environment) and, if
relevant, abstract or symbolic properties (function, role, meaning).
- Focus on essential and transferable aspects that help distinguish this class
from others.
- If no abstract meaning is appropriate, only provide objective descriptions.

### JSON Output Format
Return ONLY a valid JSON array of strings, without any additional text. For
example:
[
  "short description 1",
  "short description 2",
  "short description 3"
]
```

Figure 7: Prompts used for querying the LLM to generate fine-grained class descriptions.

## D  MTIL BENCHMARK AND EVALUATION METRICS

**Benchmark setup.** The MTIL benchmark evaluates order robustness by defining two distinct task sequences. **Order-I** progresses as: Aircraft → Caltech101 → CIFAR100 → DTD → EuroSAT → Flowers → Food → MNIST → OxfordPet → StanfordCars → SUN397. **Order-II** instead follows: StanfordCars → Food → MNIST → OxfordPet → Flowers → SUN397 → Aircraft → Caltech101 → DTD → EuroSAT → CIFAR100. In addition, a **Few-Shot (FS)** variant restricts training to only 16 labeled samples per class Tang et al. (2024), while preserving the same task orders as above. Together, these configurations impose diverse semantic and domain shifts, enabling a comprehensive assessment of order sensitivity.

**Evaluation metrics.** MTIL adopts three standard measures: *Transfer*, *Average*, and *Last*. Let $a_j^i$ denote the accuracy on task $j$ after completing task $i$, with $N$ tasks in total and $T$ the index of the final task. The metrics are defined as:

$$
\begin{aligned}
\text{Transfer} &= \frac{1}{j-1} \sum\nolimits_{i=1}^{j-1} a_j^i, \quad j = 2, \dots, N, \\
\text{Average} &= \frac{1}{N} \sum\nolimits_{j=1}^{N} a_j^T, \\
\text{Last} &= a_j^T, \quad j = 1, \dots, N.
\end{aligned}
\tag{19}
$$

Here, *Transfer* evaluates zero-shot forward generalization, *Last* reflects the retention of earlier tasks after the final training step, and *Average* provides an overall summary of continual learning performance across all tasks. A robust algorithm should maintain stable results under all task orders and data regimes.

# E  TRAINING AND TESTING ALGORITHMS

## E.1  TRAINING PROCEDURE

---

**Algorithm 1** Training process of CoFiCL

---

**Require:** Task sequence $\{D^1, \ldots, D^T\}$, pretrained image encoder $f_i$, text encoder $f_t$, prompt pool $\mathcal{P}$, Top-$K$, loss weights $\lambda_{\text{coarse}}, \lambda_{\text{proto}}, \lambda_{\text{sim}}$
**Ensure:** Adapters $\{\mathcal{A}^t\}$, Conceptual Prototypes $\{p_n\}$, prompt counts $\{C_m\}$
 1: **for** $t = 1, \ldots, T$ **do**
 2:    Construct coarse prompts $t_c^{\text{coarse}}$ for task $t$
 3:    Construct fine-grained prompts from LLM if applicable
 4:    Initialize adapter $\mathcal{A}^t$ and freeze previous adapters
 5:    **for all** minibatch $(\mathbf{x}, y)$ in $\mathcal{T}^t$ **do**
 6:       Compute image features $z_{\text{cls}} = f_i(\mathbf{x})$
 7:       **Coarse branch:** pass through $\mathcal{A}^t$, compute coarse logits, loss $\mathcal{L}_{\text{coarse}}$
 8:       **Fine branch:** query prompt pool $\mathcal{P}$ using $z_{\text{cls}}$, select top-$K$ prompts
 9:       Prepend selected prompts, pass through image encoder to get prompt-modulated features $\tilde{z}_i$
10:       Align $\tilde{z}_i$ with concept prototypes, compute prototype loss $\mathcal{L}_{\text{proto}}$
11:       Update prompt selection counts $C_m$, compute prompt similarity loss $\mathcal{L}_{\text{sim}}$
12:       Update adapter and prompt parameters using total loss:

$$\mathcal{L} = \lambda_{\text{coarse}}\mathcal{L}_{\text{coarse}} + \lambda_{\text{proto}}\mathcal{L}_{\text{proto}} + \lambda_{\text{sim}}\mathcal{L}_{\text{sim}}$$

13:    **end for**
14:    Update concept prototypes $\{p_n\}$ using fine prompt embeddings
15: **end for**

---

## E.2  TESTING PROCEDURE

---

**Algorithm 2** Testing process of CoFiCL

---

**Require:** Test image $x$, frozen encoders $(f_i, f_t)$, adapters $\{\mathcal{A}^t\}$, Conceptual Prototypes $\{p_n\}$, prompt pool $\mathcal{P}$, Top-$K$, balance weight $\alpha$
**Ensure:** Predicted class $\hat{y}$
 1: Extract image feature $z_q = f_i(x)$
 2: Compute coarse logits for all adapters and fuse via weighted sum (optional Gaussian scaling if available)
 3: Query prompt pool $\mathcal{P}$ with $z_q$, select top-$K$ prompts
 4: Prepend prompts, pass through encoder to get $\tilde{z}_q$
 5: Compute cosine similarity between $\tilde{z}_q$ and concept prototypes, obtain fine logits
 6: Fuse coarse and fine logits:
$$s(x) = \alpha \cdot s^{\text{coarse}}(x) + (1 - \alpha) \cdot s^{\text{fine}}(x)$$

 7: Predict class: $\hat{y} = \arg\max_c s_c(x)$
 8: **return** $\hat{y}$

---

# F  ADDITIONAL EXPERIMENTAL RESULTS

## F.1  FULL RESULTS ON ORDER-II AND FEW-SHOT SETTINGS

The complete results for MTIL Order-II and Few-Shot (FS) settings are provided in Table 6 and Table 7, respectively. Each table reports the three standard metrics (*Transfer*, *Average*, *Last*) for a fair comparison under its respective experimental setup.

## F.2  PER-TASK ACCURACY ACROSS ALL DATASETS

To provide a more detailed view of the performance of CoFiCl, Tables 8, 9, and 10 report the per-task accuracy of our method across all 11 datasets in the MTIL benchmark under three settings: Order-I, Order-II, and Few-Shot (FS).

Table 6: *Transfer*, *Average*, and *Last* scores of different continual learning methods on MTIL benchmark with Order-II.

| Method | Cars | Food | MNIST | OxfordPet | Flowers | SUN397 | Aircraft | Caltech101 | DTD | EuroSAT | CIFAR100 | Average |
|---|---|---|---|---|---|---|---|---|---|---|---|---|
| Zero-shot | 24.8 | 92.9 | 68.4 | 43.8 | 47.7 | 71.4 | 85.8 | 59.5 | 89.1 | 65.8 | 62.6 | 64.7 |
| Upper Bound | 62.0 | 96.2 | 89.6 | 79.5 | 98.9 | 97.5 | 92.7 | 99.6 | 94.7 | 89.6 | 81.8 | 89.3 |
| **Transfer** | | | | | | | | | | | | |
| LwF | | 87.8 | 58.5 | 71.9 | 46.6 | 57.3 | 12.8 | 81.4 | 34.5 | 34.5 | 46.8 | 53.2 |
| iCaRL | | 86.1 | 51.8 | 67.6 | 50.4 | 57.9 | 11.0 | 72.3 | 31.2 | 32.7 | 48.1 | 50.9 |
| ZSCL | | 88.3 | 57.5 | 84.7 | 68.1 | 64.8 | 21.1 | 88.2 | 45.3 | **55.2** | 68.2 | 64.2 |
| L2P | | 70.6 | 30.7 | 78.3 | 42.8 | 38.3 | 17.4 | 75.3 | 27.4 | 23.1 | 20.7 | 42.5 |
| DualPrompt | | 79.9 | 46.9 | 85.2 | 51.3 | 45.1 | 9.3 | 82.7 | 29.9 | 42.9 | 47.2 | 52.1 |
| S-Prompts | | 59.8 | 46.2 | 67.7 | 47.5 | 43.8 | 13.5 | 76.8 | 31.4 | 22.6 | 43.5 | 45.3 |
| MoE-Adapter | | **88.8** | 59.5 | 89.1 | 69.9 | 64.4 | 18.1 | 86.9 | 43.7 | 54.6 | 68.2 | 64.3 |
| DIKI | | 85.8 | **59.8** | 89.1 | 71.8 | 62.6 | 24.3 | 93.3 | 42.7 | 46.8 | 67.8 | 64.4 |
| CoFiCL | | 86.0 | 59.3 | **89.6** | **73.3** | **65.4** | **25.4** | **93.9** | **50.9** | 50.2 | **69.6** | **66.3** |
| **Average** | | | | | | | | | | | | |
| LwF | 49.0 | 77.0 | **92.1** | 85.9 | 66.5 | 67.2 | 20.9 | 84.7 | 44.6 | 45.5 | 50.5 | 62.2 |
| iCaRL | 52.0 | 75.9 | 77.4 | 74.6 | 58.4 | 59.3 | 11.7 | 79.6 | 42.1 | 43.2 | 51.7 | 56.9 |
| ZSCL | 81.7 | **91.3** | 91.1 | 91.0 | 82.9 | **72.5** | 33.6 | 89.7 | 53.3 | **62.8** | 69.9 | 74.5 |
| L2P | 80.1 | 87.4 | 86.7 | 89.6 | 76.8 | 59.1 | 27.7 | 79.5 | 39.9 | 34.6 | 26.5 | 62.5 |
| DualPrompt | 78.6 | 88.4 | 89.7 | 91.7 | 80.0 | 62.4 | 23.2 | 85.0 | 41.3 | 51.6 | 50.7 | 67.5 |
| S-Prompts | 79.2 | 86.5 | 89.5 | 87.0 | 78.2 | 61.5 | 25.5 | 83.6 | 41.9 | 36.3 | 47.2 | 65.1 |
| MoE-Adapter | **84.9** | 89.9 | 89.3 | 91.4 | 86.2 | 72.2 | 33.4 | 89.4 | 53.3 | 61.4 | 69.9 | 74.7 |
| DIKI | 81.9 | 88.9 | **92.1** | 92.8 | 87.7 | 70.3 | 34.3 | 94.2 | 51.5 | 56.1 | 69.5 | 74.5 |
| CoFiCL | 81.8 | 89.2 | **92.1** | **93.2** | **88.2** | 71.8 | **34.6** | **94.6** | **57.5** | 58.8 | **71.2** | **75.7** |
| **Last** | | | | | | | | | | | | |
| LwF | 34.6 | 69.6 | 99.3 | 88.7 | 61.1 | 72.5 | 32.5 | 88.1 | 65.6 | 90.9 | **87.9** | 71.9 |
| iCaRL | 46.0 | 81.5 | 91.3 | 82.8 | 66.5 | 72.2 | 16.3 | 91.6 | 68.1 | 83.2 | 87.8 | 71.6 |
| ZSCL | 78.2 | **91.1** | 97.6 | 92.5 | 87.4 | **78.2** | 45.0 | 92.3 | 72.7 | 96.2 | 86.3 | 83.4 |
| L2P | 80.1 | 89.1 | 99.1 | 93.8 | 96.2 | 76.5 | 40.1 | 86.9 | 73.5 | 86.3 | 84.2 | 82.3 |
| DualPrompt | 78.6 | 89.3 | 99.2 | 94.1 | 96.5 | 76.8 | 39.8 | 89.0 | 71.6 | 90.7 | 84.9 | 82.8 |
| S-Prompts | 79.2 | 89.1 | 99.1 | 94.3 | 95.8 | 76.3 | 39.9 | 95.5 | 70.1 | 97.6 | 84.4 | 83.8 |
| MoE-Adapter | **84.1** | 88.5 | 94.0 | 91.8 | 94.1 | 77.8 | **50.4** | 93.3 | 77.1 | 87.7 | 86.6 | 84.1 |
| DIKI | 81.9 | 89.2 | **99.4** | 94.3 | **96.8** | 76.7 | 46.3 | **95.9** | 74.8 | **98.3** | 86.6 | **85.5** |
| CoFiCL | 81.8 | 89.5 | **99.4** | 94.5 | 96.7 | 77.2 | 45.8 | 95.8 | 75.2 | 97.5 | 86.4 | 85.4 |

Table 7: *Transfer*, *Average*, and *Last* scores of different continual learning methods on MTIL-FS benchmark.

| Method | Aircraft | Caltech101 | CIFAR100 | DTD | Flowers | Food | Cars | SUN397 | Average |
|---|---|---|---|---|---|---|---|---|---|
| Zero-shot | 24.8 | 92.9 | 68.4 | 43.8 | 71.4 | 85.8 | 65.8 | 62.6 | 64.4 |
| Upper Bound | 62.0 | 96.2 | 89.6 | 79.5 | 97.5 | 92.7 | 89.6 | 81.8 | 86.1 |
| **Transfer** | | | | | | | | | |
| ZSCL | | 87.3 | 67.7 | 45.4 | 67.8 | **86.6** | 59.7 | 63.4 | 68.3 |
| L2P | | 66.7 | 54.3 | 30.6 | 47.3 | 71.5 | 54.6 | 52.4 | 53.9 |
| DualPrompt | | 78.8 | 64.4 | 32.0 | 51.7 | 77.5 | 49.4 | 51.3 | 57.9 |
| S-Prompts | | 70.3 | 52.7 | 31.5 | 54.8 | 74.0 | 55.4 | 50.0 | 55.5 |
| DIKI | | 92.7 | 68.8 | 44.1 | 70.0 | 86.2 | 65.1 | 65.5 | 70.3 |
| CoFiCL | | **93.8** | **69.6** | **51.9** | **72.3** | 86.2 | **66.7** | **66.2** | **72.4** |
| **Average** | | | | | | | | | |
| ZSCL | 33.5 | 90.5 | 74.7 | 58.5 | 79.7 | **87.7** | 64.8 | 64.8 | 69.3 |
| L2P | 30.2 | 84.5 | 70.1 | 51.9 | 69.6 | 77.1 | 60.0 | 55.2 | 62.3 |
| DualPrompt | 36.5 | 89.5 | 72.5 | 52.7 | 72.3 | 80.8 | 56.1 | 54.2 | 64.3 |
| S-Prompts | 30.6 | 86.8 | 70.0 | 51.7 | 74.3 | 78.5 | 60.7 | 53.0 | 63.2 |
| DIKI | 41.3 | 95.3 | 76.5 | 58.5 | 82.2 | 86.4 | 68.2 | 66.6 | 71.9 |
| CoFiCL | **44.1** | **95.5** | **76.8** | **61.9** | **83.6** | 86.6 | **70.1** | **67.4** | **73.2** |
| **Last** | | | | | | | | | |
| ZSCL | 27.7 | 90.9 | 74.4 | 64.7 | 90.2 | **89.2** | **80.6** | 74.6 | 74.0 |
| L2P | 30.2 | 87.1 | 75.4 | 64.7 | 91.9 | 86.4 | 76.1 | 74.7 | 73.3 |
| DualPrompt | 36.5 | 91.0 | 75.1 | 65.1 | 92.9 | 86.2 | 76.2 | 74.2 | 74.7 |
| S-Prompts | 30.6 | 89.2 | 75.8 | 63.8 | 93.9 | 86.2 | 76.7 | 73.9 | 73.8 |
| DIKI | 41.3 | **95.6** | **79.0** | **67.3** | 94.4 | 86.8 | 77.6 | 74.4 | 77.1 |
| CoFiCL | **44.3** | 95.7 | 79.2 | 66.6 | **95.0** | 87.3 | 80.3 | **75.2** | **78.0** |

Table 8: Accuracy of CoFiCL on the MTIL benchmark with order-I. Each row indicates the model's performance on all datasets after learning the corresponding task. Transfer , Average , and Last metrics are highlighted in color.

| | Aircraft | Caltech101 | CIFAR100 | DTD | EuroSAT | Flowers | Food | MNIST | OxfordPet | Cars | SUN397 | |
|---|---|---|---|---|---|---|---|---|---|---|---|---|
| Transfer | | 96.47 | 88.41 | 73.05 | 98.31 | 98.05 | 89.02 | 99.42 | 94.36 | 84.37 | 66.96 | 70.2 |
| Aircraft | 51.73 | 90.39 | 69.27 | 51.06 | 51.4 | 73.53 | 86.0 | 59.88 | 89.02 | 66.7 | 65.15 | |
| Caltech101 | 51.73 | 96.39 | 68.89 | 52.07 | 51.51 | 68.66 | 85.71 | 56.92 | 88.96 | 66.2 | 66.96 | |
| CIFAR100 | 51.73 | 96.39 | 88.41 | 52.07 | 49.2 | 68.66 | 85.71 | 65.27 | 88.96 | 66.2 | 66.97 | |
| DTD | 51.73 | 96.43 | 88.41 | 73.11 | 49.09 | 67.44 | 85.36 | 65.27 | 88.96 | 66.2 | 66.96 | |
| EuroSAT | 51.73 | 96.43 | 88.41 | 73.11 | 98.31 | 67.44 | 85.36 | 65.27 | 88.96 | 66.2 | 66.96 | |
| Flowers | 51.73 | 96.43 | 88.41 | 73.11 | 98.31 | 98.05 | 85.36 | 65.27 | 88.96 | 66.2 | 66.96 | |
| Food | 51.73 | 96.43 | 88.41 | 73.05 | 98.31 | 98.05 | 89.02 | 65.27 | 88.96 | 66.2 | 66.95 | |
| MNIST | 51.73 | 96.43 | 88.41 | 73.05 | 98.31 | 98.05 | 89.02 | 99.42 | 88.96 | 66.2 | 66.95 | |
| OxfordPet | 51.73 | 96.47 | 88.41 | 73.05 | 98.31 | 98.05 | 89.02 | 99.42 | 94.36 | 66.2 | 66.95 | |
| Cars | 51.73 | 96.47 | 88.41 | 73.05 | 98.31 | 98.05 | 89.02 | 99.42 | 94.36 | 84.37 | 66.96 | |
| SUN397 | 51.79 | 96.43 | 88.41 | 73.05 | 98.31 | 98.05 | 89.01 | 99.42 | 94.36 | 84.36 | 78.37 | 86.5 |
| Average | 51.7 | 95.9 | 84.9 | 67.3 | 80.8 | 84.9 | 87.1 | 76.4 | 90.4 | 69.5 | 67.8 | 77.9 |

Table 9: Accuracy of CoFiCL on the MTIL benchmark with order-II. Each row indicates the model's performance on all datasets after learning the corresponding task. Transfer , Average , and Last metrics are highlighted in color.

| | Aircraft | Caltech101 | CIFAR100 | DTD | EuroSAT | Flowers | Food | MNIST | OxfordPet | Cars | SUN397 | |
|---|---|---|---|---|---|---|---|---|---|---|---|---|
| Transfer | | 89.54 | 99.39 | 94.52 | 96.67 | 77.17 | 45.81 | 95.82 | 75.24 | 97.49 | 69.27 | 66.3 |
| Aircraft | 81.84 | 85.99 | 62.12 | 88.99 | 73.53 | 65.22 | 25.68 | 93.59 | 51.3 | 51.53 | 69.3 | |
| Caltech101 | 81.84 | 89.53 | 56.4 | 89.86 | 73.2 | 65.47 | 25.68 | 93.83 | 50.77 | 52.51 | 70.14 | |
| CIFAR100 | 81.84 | 89.53 | 99.38 | 89.86 | 73.2 | 65.47 | 25.68 | 93.83 | 50.77 | 52.51 | 70.14 | |
| DTD | 81.84 | 89.53 | 99.38 | 94.44 | 73.28 | 65.44 | 25.68 | 93.79 | 50.83 | 52.51 | 70.13 | |
| EuroSAT | 81.84 | 89.53 | 99.38 | 94.44 | 96.55 | 65.44 | 25.68 | 93.91 | 50.89 | 52.51 | 70.14 | |
| Flowers | 81.84 | 89.54 | 99.38 | 94.52 | 96.67 | 77.17 | 23.7 | 94.08 | 50.71 | 47.52 | 69.32 | |
| Food | 81.84 | 89.54 | 99.38 | 94.52 | 96.67 | 77.17 | 45.75 | 94.08 | 50.71 | 47.52 | 69.32 | |
| MNIST | 81.84 | 89.54 | 99.38 | 94.52 | 96.67 | 77.17 | 45.81 | 95.82 | 50.89 | 47.53 | 69.42 | |
| OxfordPet | 81.84 | 89.54 | 99.38 | 94.52 | 96.67 | 77.17 | 45.81 | 95.82 | 75.24 | 47.93 | 69.25 | |
| Cars | 81.84 | 89.54 | 99.38 | 94.52 | 96.67 | 77.17 | 45.81 | 95.82 | 75.24 | 97.49 | 69.27 | |
| SUN397 | 81.84 | 89.54 | 99.39 | 94.52 | 96.67 | 77.17 | 45.81 | 95.82 | 75.24 | 97.49 | 86.4 | 85.4 |
| Average | 81.8 | 89.2 | 92.1 | 93.2 | 88.2 | 71.8 | 34.6 | 94.6 | 57.5 | 58.8 | 71.2 | 75.7 |

Table 10: Accuracy of CoFiCL on the MTIL-FS benchmark. Each row indicates the model's performance on all datasets after learning the corresponding task. Transfer , Average , and Last metrics are highlighted in color.

| | Aircraft | Caltech101 | CIFAR100 | DTD | Flowers | Food | Cars | SUN397 | |
|---|---|---|---|---|---|---|---|---|---|
| Transfer | | 93.8 | 69.6 | 51.9 | 72.3 | 86.2 | 66.7 | 66.2 | 72.4 |
| Aircraft | 44.1 | 93.79 | 69.19 | 51.24 | 73.49 | 86.02 | 66.71 | 65.12 | |
| Caltech101 | 44.04 | 95.74 | 70.04 | 52.25 | 71.86 | 86.21 | 66.72 | 66.42 | |
| CIFAR100 | 44.04 | 95.74 | 79.18 | 52.25 | 71.86 | 86.21 | 66.72 | 66.42 | |
| DTD | 44.04 | 95.74 | 79.18 | 68.09 | 71.86 | 86.21 | 66.72 | 66.42 | |
| Flowers | 44.04 | 95.74 | 79.18 | 68.2 | 95.01 | 86.21 | 66.72 | 66.42 | |
| Food | 44.04 | 95.74 | 79.18 | 68.14 | 95.01 | 87.31 | 66.72 | 66.43 | |
| Cars | 44.04 | 95.74 | 79.18 | 68.14 | 95.01 | 87.31 | 80.26 | 66.42 | |
| SUN397 | 44.34 | 95.7 | 79.21 | 66.61 | 95.01 | 87.32 | 80.31 | 75.16 | 78.0 |
| Average | 44.1 | 95.5 | 76.8 | 61.9 | 83.6 | 86.6 | 70.1 | 67.4 | 73.2 |