# OpenReview forum: "CoFiCL: Coarse-to-Fine Continual Learning with Hierarchical Knowledge"
_ICLR.cc/2026/Conference — Submitted to ICLR 2026_

### Official Review · Reviewer_Y4m1 · 2025-10-21

**Soundness:** 2
**Presentation:** 2
**Contribution:** 2
**Rating:** 4
**Confidence:** 4

**Summary:**

This paper tackles catastrophic forgetting in CIL using VLMs like CLIP, finding that simple category prompts ignore the fine grained knowledge that could enrich semantic representations. Therefore, the authors construct fine-grained prompts via LLMs, and use them to further guide the training of the adapters.

**Strengths:**

1.	The illustration is clear.
2.	The proposed method using fine-grained descriptions of the class is reasonable.

**Weaknesses:**

1.	Figure 1 does not convey useful information. The samples in Figure 1b are not discriminative. Also, I don’t think t-SNE visualization supports the conclusion, because t-SNE groups the similar samples, and the visualization is affected by the number of iteration steps. For example, the conclusion that “category prompts (stars) and concept prompts (circles) remain separated” in (c) cannot be concluded because (c) in Figure 1 clearly uses a smaller number of iterations than (a).
2.	There are no Related Works section in the main text of this paper, making it unclear that how this paper related to other papers in the area.
3. The “coarse-fine path” structure proposed by CoFiCL, which uses task-specific adapters and prototype-based contrastive learning to handle coarse and fine-grained knowledge, is not novel in the context of incremental learning. Many existing studies have already adopted similar strategies, such as prompt learning [1, 2, 3], incorporating adapters [4], prototype learning [5, 6], and the aid of LLM [7, 8] to enhance the model’s transferability and reduce forgetting. As a result, CoFiCL lacks sufficient innovation.
4.	CoFiCL proposes the use of LLMs to generate text descriptions that complement “A photo of a [class].” On one hand, this is not a new idea [7, 8]; on the other, it represents the main contribution of the paper. However, the biggest challenge in the CIL domain is catastrophic forgetting, and the authors have merely applied well-established Gaussian replay methods [5, 6] without making any notable contribution in this area.

[1] Dualprompt: Complementary prompting for rehearsal-free continual learning. ECCV22
[2] Coda-prompt: Continual decomposed attention-based prompting for rehearsal-free continual learning. CVPR23
[3] Learning to prompt for continual learning. CVPR22
[4] Class-Incremental Learning with CLIP: Adaptive Representation Adjustment and Parameter Fusion. ECCV24
[5] Prototype augmentation and self-supervision for incremental learning. CVPR21
[6] SLCA: Slow Learner with Classifier Alignment for Continual Learning on a Pre-trained Model. ICCV23
[7] ChatGPT-Powered Hierarchical Comparisons for Image Classification. NeurIPS23
[8] Tree of attributes prompt learning for vision-language models. ICLR25

**Questions:**

1.	How does the fine-grained knowledge relate to the mitigation of catastrophic forgetting? It helps the classification, but seems not directly related to the incremental learning.

---

### Official Review · Reviewer_aXZW · 2025-10-22

**Soundness:** 2
**Presentation:** 2
**Contribution:** 2
**Rating:** 2
**Confidence:** 4

**Summary:**

This paper proposes a dual-path continual learning with CLIP for image classification, where texts of class name and more detailed class properties are respectively used in the two paths for better learning of visual classes. Initial experiments on one multi-task benchmark supports the efficacy of the proposed method.

**Strengths:**

The proposed dual-path framework based on coarse and fine conceptual knowledge are novel and potentially useful for learning of image classes, and has been initially confirmed by the experiments.

**Weaknesses:**

1.	The proposed method is probably effective only for the domain-class incremental learning (DCIL) and therefore its application is very limited. Considering both classes and domains are different between tasks in DCIL, such continual learning setting is actually a much simpler one compared to widely used class-incremental learning (CIL). The core technique (Gaussian centers and prompt prototypes) used in the proposed method are likely only valid for DCIL but not for CIL, because classes across tasks in CIL are often similar to each other such that the task selection strategy in the proposed method would fail.
2.	The presentation of the essential part of the proposed method is ambiguous and probably erroneous. For the “Fine grained knowledge learning” subsection, (1) it is not clear how each prompt cluster is related to each class, how each sample is assigned to one prototype, and why minimizing L_proto can help classify visual classes; (2) it is not clear why minimizing L_sim can help avoid prompt over concentration and  select correct prompts; (3) how the weighted sum in  s^coarse and s^fine can be used to predict any class from all the learned tasks (it seems only for one task rather than for all tasks), how to compute s^coarse and s^fine, and what are the dimensionality of s^coarse and s^fine? (4) subscript of ‘s’ in Eq 2 represents one class, but subscript of ‘s’ in Eqs 16 and 17 respectively represent task/adapter and prompt, making it very confusing about the meaning and calculation of “s”.
3.	The empirical evaluations are quite limited. First, the method is evaluated only in the DCIL. Evaluations in the CIL setting should be included. Second, the baselines used in experiments are mostly out-of-date and recently proposed method in 2024 and 2025 are largely missing (only 2 baselines from 2024 and none from 2025). Third, some key ablation studies are missing. In particular, what if not using cluster center but class-specific concept prompt for the contrastive loss? Fourth, some hyper-parameter setting are missing or not discussed. For example, what is the pool size for most experiments and how to select its value? How to select prototype numbers for each task dataset and how about the sensitivity of prototype numbers?
4.	The claims of this paper is either misleading or incorrect. The paper claims the proposed method is about continual learning of vision-language model (VLM), but it is actually about continual learning of visual classes with the help of VLM; The paper claims the proposed method is evaluated on multiple benchmarks, but it is actually evaluated only on the MTIL benchmark.

**Questions:**

1.	Can you justify the proposed method is also valid in CIL?
2.	Can you clarify the issues raised from Weakness 2 above?
3.	Can you evaluate the proposed method in the more general CIL setting, and can you compare the method with more recently published methods in 2024 and 2025?

---

### Official Review · Reviewer_f7y3 · 2025-10-27

**Soundness:** 1
**Presentation:** 2
**Contribution:** 2
**Rating:** 2
**Confidence:** 4

**Summary:**

This paper introduces CoFiCL, a dual-path framework for continual learning in Vision-Language Models. It separates learning into a coarse path using task-specific adapters for category discrimination and a fine path using LLM-generated concepts to train a pool of visual prompts for better transferability. The outputs are fused for final prediction. The method is evaluated on the MTIL benchmark, where it shows performance improvements over several baselines.

**Strengths:**

1. The case study on error correction in Fig. 6 effectively demonstrates how the fine-grained concept path can correct errors made by the coarse category path, intuitively showcasing the advantages of CoFiCL.

**Weaknesses:**

1. **Comparison on Transfer Metric:** CoFiCL utilizes attribute-augmented prompts generated by an LLM, which provide richer descriptive information for each class. However, such enriched prompts can boost the zero-shot capabilities of CLIP. The AWT [1] method mentioned in the paper reports an average zero-shot performance increase of 6.5% across 14 datasets using this technique in Tab. 1. However, all baseline methods compared in this paper, as well as the reported zero-shot CLIP baseline, are based on the simple "a photo of a {class name}" template. This discrepancy in prompt engineering gives CoFiCL an inherent and potentially advantage in zero-shot transfer that is unrelated to its architectural contributions for continual learning.
2. **Limited Novelty:** The coarse path architecture is nearly identical to DIKI [2], while the fine path's core idea of using attribute-augmented prompts is borrowed from AWT [1]. The combination of these existing techniques appears straightforward.
3. **Parameter Count:** CoFiCL uses 27.3M tunable parameters, over 15 times more than DIKI [2] with only 1.8M, as shown in Table 2(a). This vast difference in model capacity makes it unclear if performance gains stem from the proposed architecture or simply from having more tuning parameters. I would suggest the authors conduct a controlled experiment where the parameter count of the DIKI baseline is increased to a comparable level to provide a fairer comparison.
4. **Computational Cost:** The dual-path design is likely much more expensive than single-path methods, a fact corroborated by the reported use of an A100 (80G) GPU versus a NVIDIA 3090 (24G) for DIKI. An analysis of training and inference time or FLOPs is essential for assessing practical feasibility.
5. **Writing Issues:** Vague references like "as detailed in Appendix" should point to specific sections. The ablation study (Tab. 3) would be more clear if each line is explicitly linked to the methodological component being ablated.

[1] Zhu Y, Ji Y, Zhao Z, et al. Awt: Transferring vision-language models via augmentation, weighting, and transportation[J]. Advances in Neural Information Processing Systems.

[2] Tang L, Tian Z, Li K, et al. Mind the interference: Retaining pre-trained knowledge in parameter efficient continual learning of vision-language models[C]//European conference on computer vision.

**Questions:**

1. The statement in the introduction, "the large language model generates multiple concept descriptions without explicit class names, for example It is {desc}. These prompts are only applied on the visual side." is unclear. Could the authors please elaborate how text descriptions are "applied on the visual side"?
2. The L_sim loss (Eq. 12) appears to favor prompts with higher historical usage frequency, potentially leading to reduced prompt diversity (a rich-get-richer scenario). Could you justify this design choice, which seems contrary to load balancing design in MoE [3]? Also, does the ablation in Line 6 of Tab. 3 correspond to the removal of this mechanism?

[3] Fedus W, Zoph B, Shazeer N. Switch transformers: Scaling to trillion parameter models with simple and efficient sparsity[J]. Journal of Machine Learning Research, 2022.

---

### Official Review · Reviewer_jiJF · 2025-10-31

**Soundness:** 3
**Presentation:** 3
**Contribution:** 3
**Rating:** 4
**Confidence:** 4

**Summary:**

The paper proposes CoFiCL, a coarse-to-fine continual learning framework for CLIP-style vision-language models in domain-class incremental learning (DCIL). The method splits semantic modeling into two parallel paths: (1) a coarse path that preserves category-discriminative knowledge via task-specific adapters and aligns to category prompts, using Gaussian task centers for adapter selection/weighting; (2) a fine path that uses LLM-generated class-agnostic concept descriptions, a learnable retrievable visual prompt pool, and prototype-based contrastive alignment between prompt-modulated image features and text-space concept prototypes. Experiments on the MTIL benchmark (Order I, Order II, Few-Shot) show improvements over several baselines on Transfer, Average, and Last metrics.

**Strengths:**

1. The design of the proposed method is intuitive and interpretable, with a clean separation into coarse (category-level) and fine (concept-level) paths that align with the geometry of prompt embeddings.

2. Experiments on the large multi-domain MTIL benchmark demonstrate consistent improvements across various settings compared to baselines.

**Weaknesses:**

1. Limited novelty: Most building blocks have prior art (adapters, prompt pools, LLM-generated attributes, prototype contrastive losses, top-k prompt selection, router ideas). The main novelty is in the particular coarse/fine decomposition and their fusion; the paper should more clearly delineate which elements are new versus adapted and demonstrate that the combination yields synergistic gains beyond simple additive effects.
2. Dependence on External LLM: The method relies on a proprietary, powerful LLM (GPT-4o) to generate fine-grained concepts. This creates a cost, reproducibility, and potential bottleneck concern that is not explored.
3. The paper uses k-means on text embeddings to obtain prototypes, but details are missing: are prototypes computed once offline, re-clustered per task, or updated online? How many prototypes were used and why? This matters in the incremental setup.
4. The choice of loss weights lacks justification, as the paper doesn’t provide sensitivity analysis or explain how these values were set. Authors should show how sensitive results are to these choices and provide rationale or automated schemes.
5. The approach retains one adapter per task and a growing prompt pool; moreover, the fusion formula implies computing logits for multiple adapters unless a top-K selection is used. The manuscript lacks measurements of how parameter count, memory, and inference latency scale with the number of tasks T and the effect of selection parameters (top-K). For practical continual learning deployments this is critical.
6. I'm confused because Figure 2 shows adapters placed after the encoders, while the text and implementation suggest adapters are inserted into every Transformer layer—could the authors please clarify which is correct?
7. Clarity on "Forward Forgetting": The introduction states that VLMs suffer from "forward forgetting," where pre-trained knowledge is eroded. While the results show strong forward transfer, it is less clear how the method specifically measures or mitigates the erosion of the original CLIP zero-shot capabilities on tasks outside the incremental sequence. The "Transfer" metric measures generalization to future tasks within the sequence, which is related but not identical to preserving the original pre-trained knowledge.

**Questions:**

please see the weakness

---

### Meta-Review · Area_Chair_v9ok · 2025-12-29

**Summary:**

All reviewers show to reject this paper due to the limited novelty and insufficient comparison. Moreover, the authors have not provided the rebuttal. The Meta reviewer carefully read this paper and the comments, which indeed contain many problems that remain unsolved, like the limited novelty, incomplete comparison, and overstated contributions. After considering the reviews, rebuttal, and the author's message, the Meta reviewer agrees with the concerns raised by the reviewers and recommends rejecting the paper.

**Reviewer Concerns:**

All concerns raised by the reviewers have not been addressed.

**Reviewer Scores:**

None

---

### Decision · Program_Chairs · 2026-01-26

Reject